# MicroRNA-22 Is a Key Regulator of Lipid and Metabolic Homeostasis

**DOI:** 10.3390/ijms241612870

**Published:** 2023-08-17

**Authors:** Riccardo Panella, Andreas Petri, Bhavna N. Desai, Sharmila Fagoonee, Cody A. Cotton, Piercen K. Nguyen, Eric M. Lundin, Alexandre Wagshal, Da-Zhi Wang, Anders M. Näär, Ioannis S. Vlachos, Eleftheria Maratos-Flier, Fiorella Altruda, Sakari Kauppinen, Pier Paolo Pandolfi

**Affiliations:** 1Center for Genomic Medicine, Desert Research Institute, Reno, NV 89512, USA; riccardop@dcm.aau.dk (R.P.);; 2Cancer Research Institute, Beth Israel Deaconess Cancer Center, Departments of Medicine and Pathology, Beth Israel Deaconess Medical Center, Harvard Medical School, Boston, MA 02215, USA; 3Center for RNA Medicine, Department of Clinical Medicine, Aalborg University, DK-2450 Copenhagen SV, Denmark; 4Division of Endocrinology and Metabolism, Beth Israel Deaconess Medical Center, 330 Brookline Ave, Center for Life Sciences, Boston, MA 02215, USA; 5Institute of Biostructure and Bioimaging (CNR) c/o Molecular Biotechnology Center, 10126 Turin, Italy; 6Massachusetts General Hospital Cancer Center, Department of Cell Biology, Harvard Medical School, Boston, MA 02215, USA; 7Boston Children’s Hospital, Boston, MA 02215, USA; 8Cancer Research Institute, Harvard Medical School Initiative for RNA Medicine, Department of Pathology, Beth Israel Deaconess Medical Center, Harvard Medical School, Boston, MA 02115, USA; 9Broad Institute of MIT and Harvard, Cambridge, MA 02142, USA; 10Department of Molecular Biotechnology and Health Sciences, Molecular Biotechnology Center, University of Turin, 10126 Turin, Italy; 11Renown Institute for Cancer, Nevada System of Higher Education, Reno, NV 89502, USA

**Keywords:** microRNA, drug development, non-coding RNA, Metastatic tumor, RNA medicine, therapies

## Abstract

Obesity is a growing public health problem associated with increased risk of type 2 diabetes, cardiovascular disease, nonalcoholic fatty liver disease (NAFLD) and cancer. Here, we identify microRNA-22 (miR-22) as an essential rheostat involved in the control of lipid and energy homeostasis as well as the onset and maintenance of obesity. We demonstrate through knockout and transgenic mouse models that miR-22 loss-of-function protects against obesity and hepatic steatosis, while its overexpression promotes both phenotypes even when mice are fed a regular chow diet. Mechanistically, we show that miR-22 controls multiple pathways related to lipid biogenesis and differentiation. Importantly, genetic ablation of miR-22 favors metabolic rewiring towards higher energy expenditure and browning of white adipose tissue, suggesting that modulation of miR-22 could represent a viable therapeutic strategy for treatment of obesity and other metabolic disorders.

## 1. Introduction

Obesity is considered by the WHO as the primary preventable cause of death and the fifth leading cause of death in western countries. Nearly 40% of adult Americans and Europeans [1] were described as overweight in 2017, and about 13% of the entire world’s adult population was considered obese, numbers that have nearly tripled since 1975 [2]. The epidemic of childhood obesity [3] is particularly worrisome, since in 2016 approximately 42 million children under the age of 5 were considered overweight and over 340 million children and adolescents under the age of 20 were either overweight or obese [2]. Additionally, morbid and often fatal obesity is a central manifestation of genetic disorders such as the Prader–Willi Syndrome [4]. Fat accumulation depends on the balance between anabolic processes such as adipogenesis and catabolic processes such as thermogenesis [5]. It is widely acknowledged that there are two general types of adipose tissue in mammals: white adipose tissue (WAT) and brown adipose tissue (BAT). WAT is important for energy storage and secretion of adipokines that support metabolic homeostasis, while BAT is specialized for energy expenditure and thermogenesis [6]. The balance of these processes is critical for maintaining normal adiposity and regulating lipid homeostasis.

MicroRNAs (miRNAs) are short endogenous non-coding RNAs, typically ~22 nucleotides in length, that function as post-transcriptional regulators of gene expression by repressing protein translation and promoting mRNA cleavage [7]. Previous reports provided strong evidence that miR-22 acts as an oncogene in breast cancer and leukemia by promoting metastasis and progression of myelodysplastic syndrome, respectively [8,9]. Metastatic spread is often associated with changes and perturbation in cellular metabolism [10,11], and the metabolic status and activation of lipogenic [12] programs have a strong impact on increasing the metastatic potential of certain tumors [13]. Interestingly, miR-22 is predicted to target several genes in different metabolic pathways, such as ‘fatty acid metabolism’, ‘fatty acid biosynthesis’ and ‘the central carbon metabolism pathway’ (Figure 1a), thus implicating miR-22 in the control of metabolic homeostasis. These findings are consistent with the observed metabolic advantages upon pharmacologic inhibition of miR-22 in a mouse model of diet-induced obesity [14,15]. In the present study, we deployed miR-22 knockout and transgenic mice and provide genetic evidence for an essential role of miR-22 in lipid accumulation, steatotic liver disease (STD) and obesity.

## 2. Results

To gain further insights into the role of miR-22 in metabolism, we generated an miR-22 conditional transgenic mouse model in which the miR-22 transgene (Tg) expression was controlled by Cre recombinase. Surprisingly, mice carrying the miR-22 Tg allele rapidly gained weight compared to wildtype (WT) littermates (Appendix A), likely due to the fact that the C2 locus into which the miR-22 cassette was cloned showed signs of leakiness [16,17,18]. The miR-22 transgenic mice displayed increased miR-22 levels in several tissues, such as liver, mammary fat pads and visceral fat (Appendix A). To determine the tissue in which miR-22 overexpression was driving the obese phenotype, we compared different Cre-mediated, tissue-specific miR-22 transgenic mouse lines and observed that mice with high levels of miR-22 in WAT and the liver rapidly gained weight and became obese (Appendix A). However, the most pronounced obese phenotype was developed in mice in which miR-22 overexpression was driven by the liver-specific Alb-Cre (Figure 1b,c and Appendix A). Furthermore, overexpression of miR-22 in the liver was able to induce hepatic steatosis in mice fed on chow (Figure 1d) comparable to steatosis developed in WT mice fed a high-fat diet (HFD). This implies that hepatic overexpression of miR-22 is sufficient to drive obesity in mice that are fed a normal diet and is associated with accumulation of fat in the liver. Of note, WT mice fed an HFD for 8 weeks showed elevated miR-22 levels (Appendix A) compared to mice fed on chow, implicating miR-22 in the regulation of excess body fat accumulation and obesity.

Next, we investigated the role of miR-22 in metabolism using an miR-22 knockout (miR-22^−/−^ KO) mouse model (a kind gift from Dr. Da-Zhi Wang; see Materials and Methods for details). We fed wildtype (wt) and miR-22^−/−^ mice an HFD for 8 weeks and observed that the miR-22 KO mice gained significantly less weight compared to WT mice (Figure 1e, Appendix A). Furthermore, HFD-induced hepatic steatosis was markedly reduced in KO mice compared to WT littermates (Figure 1f). Hematoxylin–Eosin (H&E) staining of WAT in miR-22 KO and WT mice (Figure 1g and Appendix A) revealed that adipocytes were profoundly reduced in size and number in the KO mice after 8 weeks on an HFD, and immunohistochemistry showed strong staining for the uncoupling protein 1 (UCP-1) in the miR-22 deficient mice (Figure 1h and Appendix A). UCP-1 is a mitochondrial marker that is usually upregulated in brown adipose tissue (BAT). Thus, the observation that WAT in miR-22^−/−^ mice stains positive for a BAT marker suggests that miR-22 deficiency induces brownization of WAT, which could at least in part explain why miR-22^−/−^ KO mice do not gain weight when fed a HFD. Of note, the fat mass was markedly increased in WT mice after 8 weeks on HFD, but showed no significant change in HFD-fed miR-22 KO mice (Appendix A), whereas genetic ablation of miR-22 did not affect the body fat upon feeding the KO mice normal chow (NC). Furthermore, the percentage of lean mass remained stable in miR-22 KO mice (Appendix A).

Next, we asked whether the lack of weight gain in miR-22^−/−^ mice was due to changes in food intake or energy expenditure. At baseline, WT and miR-22^−/−^ mice showed no differences in oxygen consumption (VO_2_), food intake, respiratory exchange ratio or locomotor activity (Figure 2a,c and Appendix A). However, after 8 weeks on HFD, the miR-22^/−^ mice had a significantly higher VO_2_ consumption compared to WT mice (Figure 2b). This effect was independent of the differences in body weight, as shown by an analysis of covariance (ANCOVA) [19] (mass effect: *p* = 0.0166; group effect: *p* < 0.001) (Figure 2d). During the entire length of the study, we observed no changes in food intake between the WT and miR-22^−/−^ mice (Appendix A), indicating that the effect of miR-22 loss of function on body weight is not related to a decrease in appetite. Data collected from metabolic cages showed no difference in respiratory exchange ratio or locomotor activity (Appendix A) between the two mouse cohorts; however, thermal camera analysis revealed that the body temperature was on average 3 degrees higher in miR-22^−/−^ mice compared to WT animals (Figure 2e and Appendix A). The thermal signal was localized in the intra-scapular area, where brown fat accumulates, and was more pronounced in miR-22^−/−^ mice compared to WT littermates, suggesting that loss of miR-22 function leads to metabolic reprogramming when the miR-22 KO mice are challenged with an HFD. In addition, embryonic fibroblasts (MEFs) isolated from miR-22^−/−^ mice were much less prone to adipocyte differentiation compared to WT cells (Figure 2f and Appendix A). Taken together, these findings imply that miR-22 is involved in fat accumulation and that its overexpression leads to an obese phenotype. By contrast, fat accumulation is impaired in miR-22 KO mice fed a HFD, likely due to metabolic rewiring resulting in increased energy expenditure and suppression of weight gain.

To investigate if miR-22-3p is associated with human obesity, we analyzed RNA sequencing (RNA-Seq) data of 86 adipose tissue needle biopsies from patients from the METSIM study with BMI > 30 (*n* = 30) or BMI < 25 (*n* = 56). In this analysis, miR-22-3p was one of the most highly expressed miRNAs, comprising ~4.5% of all miRNA reads (Figure 2h,i) [20]. Comparison of obese and normal-weight individuals showed that miR-22-3p was significantly overexpressed in obese human adipose tissue (*p*-value: 0.0078; q-value: 0.0373) (Figure 2j) [20]. Its expression also correlated positively with different anthropometric and metabolic characteristics, such as waist–hip ratio, body-mass index, serum triglycerides, serum ApoB, HOMAIR, serum alanine aminotransferase and serum C-reactive protein (Appendix A). Furthermore, miR-22-3p levels correlated negatively with muscle mass, serum adiponectin and serum HDL cholesterol (Appendix A), and several metabolic pathways (Appendix A) were predicted to be under the control of miR-22, indicating that the role of miR-22 in metabolic homeostasis is conserved in mice and humans (Figure 2g). Upon analyzing a cohort of 122 patients with different grades of liver damage, ranging from steatosis to grade 3 fibrosis (F3), we observed that miR-22 levels positively correlate with human disease progression. As shown in Figure 2k, the patients with liver fibrosis have higher levels of miR-22 compared with the sub-group with hepatic steatosis. Moreover, when the fibrosis advanced from grade 1 (F1) to grade 3, the amount of miR-22 also increased with the severity of the disease, providing further evidence of the crucial role of this microRNA in controlling biological aspects connected with liver disease progression and specifically fibrosis in NASH patients.

## 3. Discussion

In this study, we provide genetic evidence for miR-22 as a key player in the control of lipid and energy homeostasis as well as the onset and maintenance of obesity using miR-22 knockout and transgenic mice. These findings are consistent with the fact that miR-22 regulates direct mRNA targets implicated in several key metabolic pathways [14,15]. Furthermore, we show that genetic ablation of miR-22 enhances beiging of WAT, thereby increasing energy expenditure and impairing lipid biosynthesis, as showed in the schematic model presented in Figure 2l Finally, we report that miR-22 levels are upregulated in obese human subjects (Figure 2j). Taken together, these data suggest that elevated levels of miR-22 could represent a physiological mechanism favoring energy and lipid accumulation. While this mechanism may be beneficial under fasting or starvation, overexpression of miR-22 could become detrimental under consumption of a high-fat diet and high consumption of food resulting in weight gain and obesity. Our findings are also consistent with the pro-metastatic role of miR-22 [9] and with the fact that a lipogenic switch favors cancer progression and metastasis [13,21]. Importantly, our data suggest that inhibition of miR-22 function could represent a new therapeutic strategy for obesity and liver steatosis.

In a separate study, we developed an antisense oligonucleotide (antimiR-22) for targeting of miR-22 (Panella et al., in preparation). Pharmacologic inhibition of miR-22 deploying the antimiR-22 oligonucleotide validated our findings presented here using miR-22 genetic mouse models. Furthermore, additional pharmacology studies in mice enabled a more detailed dissection of the molecular mechanisms underlying the role of miR-22 in modulating lipid and energy homeostasis, which is a limitation of the present study.

The past decade has witnessed significant advances in the development of new therapeutic approaches for the treatment of obesity, ranging from bariatric surgery to pharmacological therapies designed to decrease appetite, increase satiety, and boost metabolism [22]. However, while effective, surgical treatments are associated with serious side effects [23,24,25]. Approved pharmacological approaches, which are mainly based on GLP1-receptor agonists, have provided robust effects on weight loss in obese subjects; however, there are concerns regarding maintenance of body weight due to the reported adverse effects associated with the long-term use of GLP-1 agonists [26,27]. Furthermore, therapies that reduce food consumption and appetite affect both fat and lean mass, which can cause a reduction in muscle mass with an associated series of unwanted effects. Thus, there is still a high unmet medical need for new effective and safe therapies for treatment of obesity. Here, we present genetic evidence for miR-22 as an important rheostat in controlling lipid homeostasis and the onset and maintenance of obesity. Importantly, genetic ablation of miR-22 protects mice from obesity under HFD conditions without affecting food consumption or lean mass. Taken together, our data suggest that pharmacological inhibition of miR-22 function could represent a viable therapeutic strategy for obesity and liver steatosis, with a mechanism that would be complementary to treatment with GLP1 receptor agonist drugs [22,28,29], enabling the development of novel combination therapies for more effective treatment of obesity.

## 4. Materials and Methods

### 4.1. Data Analysis

miRNA expression data from human abdominal subcutaneous fat tissue needle biopsies were obtained from the METSIM study [20]. A total of 185/200 samples passed QC for having comparable total reads on mature miRNAs (<10^6^ and >6 × 10^6^). Comparison of miRNA expression between BMI > 30 (*n* = 30) and BMI < 25 (*n* = 56) individuals was performed using DESeq2 [30] with the first two expression principal components (PCs), the second PC as a factor, sample sequencing depth and the subject’s age as covariates. Storey’s q-value [31] was calculated per gene to control the type I error rate within the experiment. miRNAs with a *q*-value < 0.05 were deemed as differentially expressed between the two groups. Correlation analysis was performed by calculating Spearman’s rho on library-size normalized data using limma [32]. The same covariates as in differential expression analysis were incorporated into the model, with subject group (>30 BMI and <25 BMI) as a treatment factor to be preserved. Storey’s q-value was calculated to control for the multiple tests conducted. Correlations with a q-value < 0.05 were deemed as statistically significant. As in the METSIM publication, Log10- and Log2-transformed values were included in the correlation analysis. In the relevant table, the results correspond to the analysis of log10 values of BMI, total triglycerides, creatinine clearance rate, serum alanine aminotransferase, serum ApoB, HDL cholesterol, HOMAIR, serum CRP, OGGT fasting plasma insulin, Matsuda Insulin Sensitivity Index, plasma adiponectin, serum apoA1 and insulin AUC. KEGG miRNA enrichment was performed using experimentally supported miRNA interactions from TarBase 8 database [33]. Fisher’s exact test was performed against Central Carbon Metabolism in Cancer pathway (hsa05230) genes.

### 4.2. Mouse Models

All experiments were carried out in miR-22^−/−^ mice (a kind gift from Da-Zhi Wang) or miR-22 Tg^+/+^/Albumin-Cre mice, obtained by crossing an miR-22 Tg mouse as described in [8,9,34] with the commercially available Alb-Cre mouse model (The Jackson Laboratory, mouse strain *B6.Cg-Speer6-ps1Tg(Alb-Cre)21Mgn/J*, Cat. #003574). The same strategy was use to generate miR+22^+/+^/MMTV-Cre. Mice were kept under a 12 h light: 12 h dark cycle and an ambient temperature of 22 ± 2 °C. The mice were given ad libitum access to normal chow or a standard high-fat diet with 60% total caloric intake from lard (TD.06414) and drinking water for the entire duration of the experiment. Mouse weight was monitored once a week using a regular scale for the duration of the experiment.

### 4.3. Indirect Calorimetry, Body Composition and Thermal Imaging

All experiments were carried out in 10-week-old female miR-22^−/−^ mice and age-matched WT siblings. Mice were kept under a 12 h light: 12 h dark cycle and an ambient temperature of 22 ± 2 °C. The mice were given ad libitum access to a standard high-fat diet and drinking water for 8 weeks. The high-fat diet had a composition of 60.3% fat, 18.4% protein and 21.3% carbohydrate (9% sucrose) (Envigo, TD.06414)

Body weights were assessed once a week in the morning, between 8:00 and 10:00. Body composition, including lean and fat mass, was assessed using an Echo MRI 3-in-1 quantitative nuclear magnetic resonance (qNMR) system (Echo Medical Systems, Houston, TX, USA). This test is performed in conscious mice that are immobilized for one minute. This test was performed at Week 0 (Baseline, pre-diet) and Week 8 (Test, after 8 weeks on the diet).

Metabolic parameters were measured by indirect calorimetry using the Comprehensive Lab Animal Monitoring System (CLAMS; Columbus Instruments, Columbus, OH, USA). This included VO_2_ consumption (a measure of energy expenditure), respiratory exchange ratio (R.E.R), energy intake and ambulatory activity. Mice were individually housed with ad libitum access to food and water. Analysis was performed at 24 °C under a 12:12 h light–dark cycle (light period 06:00–18:00). Mice were acclimated in the metabolic chambers for 48 h before collecting 4 days of measurements used for data analysis. This procedure was performed at Week 0 (Baseline, pre-diet) and Week 8 (Test, after 7 weeks on the diet), following the procedure previously described [35].

The infrared camera T420sc (emissivity of 0.98, FLiR System) was used as previously reported [36] to assess the amount of BAT in mice.

All procedures were in accordance with the National Institutes of Health Guidelines for the Care and Use of Animals and approved by the Institutional Animal Care and Use Committee at Beth Israel Deaconess Medical Center (Boston, MA, USA).

### 4.4. Tissue Harvesting and Manipulation

All tissues were harvested fresh in the first 5 min after sacrifice, washed in PBS and divided in three parts on ice. One part was used for paraffin embedding and subsequently cut and stained for the desired markers, and two portions were frozen by immersion in liquid nitrogen and preserved for future analysis.

### 4.5. H&E and HIC Staining

Tissues were fixed in 4% paraformaldehyde overnight, paraffin-embedded and then sectioned at 5 μm. After deparaffinization and rehydration, antigen retrieval was performed in a pressure cooker with sodium citrate buffer at 95 °C for 25 min. Sections were incubated in a 0.3% H_2_O_2_ solution in 1 × PBS and then in a 10% serum solution in 1 × PBS for 30 min. Each solution was used to block endogenous peroxidase and background from the secondary antibody. The sections were stained with the primary antibodies and incubated in a biotinylated secondary antibody in 1 × PBS (1:500–1:1000) at room temperature for 30 min. The Vectastain ABC Elite kit was used to enhance specific staining, and the staining was visualized using a 3′-diaminobenzidine (DAB) substrate. Stained sections were counterstained using hematoxylin and dehydrated before they were sealed with a coverslip with Richard-Allan Scientific^®^ Cytoseal™ XYL Mounting Medium and then analyzed on a Nikon Eclipse 50i microscope or Olympus BX41 equipped with an Olympus Q Color 5 camera.

### 4.6. Statistical Analysis

Statistical analyses of staining as well as mice monitoring were performed with the ANOVA tool (2-way or 3-way in accordance with the experiment) included in Prism Graph Pad 8. *p*-values used were 0.1234 (Ns), 0.0332 (*), 0.0021 (**), 0.0002 (***) and <0.0001 (****).

### 4.7. Isolation and Manipulation of Primary Mouse Cells

Mouse mesenchymal cells were isolated and cultured as previously described [37].

## Figures and Tables

**Figure 1 ijms-24-12870-f001:**
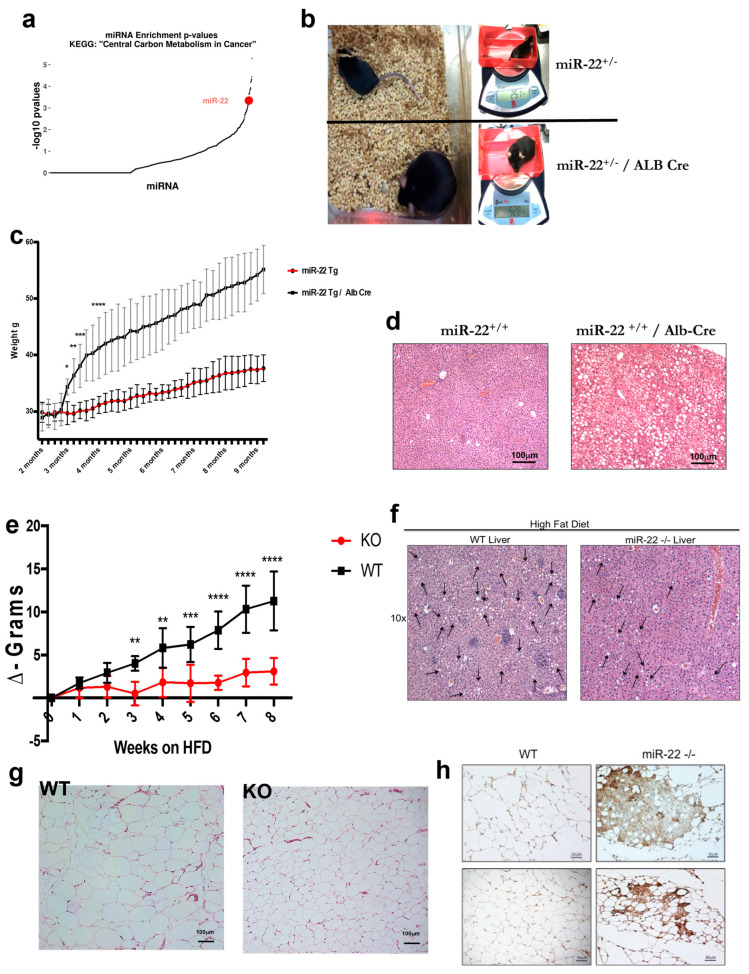
Role of miR-22 in metabolism and obesity. (**a**) The central carbon metabolism pathway is enriched in miR-22 targets. Dot plot depicting -log10 Fisher’s exact test *p*-values for all human miRNAs (*n* = 1077) in TarBase v8 database of experimentally supported miRNA interactions [29156006]. miR-22 is marked in red. (**b**) Representative comparison between miR-22^+/−^ and miR-22^+/−^/Alb-Cre littermates at 8 months of age; mice were fed with regular chow for the entire length of the experiment. miR-22Tg/Alb-Cre mice weighed over two times more than their respective littermates without Cre. (**c**) Comparison between miR-22^+/+^ transgenic mice with or without Albumin Cre expression. Mice fed with regular chow showed a striking difference between the two genotypes; miR-22^+/+^-Alb-Cre mice gained much more weight than miR-22^+/+^ and started to become obese (>40 g) around 6 months of age. The entire colony was over the obesity threshold by 9 months of age (*n* = 6 per cohort). (**d**) Livers from miR-22^+/+^ transgenic mice with or without Albumin Cre were harvested at 10 months of age. H&E staining revealed marked liver steatosis in miR-22^+/+^-Alb-Cre mice fed regular chow, while there was no sign of liver steatosis in livers from WT mice. (**e**) HFD-fed miR-22 deficient mice failed to gain weight compared to WT littermates, suggesting a role of miR-22 in diet-induced obesity (*n* = 5 per cohort). (**f**) Livers from miR-22^−/−^ mice were harvested after 8 weeks on an HFD. H&E staining revealed a strong protective effect against hepatic steatosis in the mice lacking miR-22. (**g**) After 8 weeks on an HFD, mice were sacrificed and tissues harvested. WAT from miR-22-KO and WT mice stained with H&E showed a significant difference in adipocyte size, which was much smaller in miR-22 KO mice compared to their WT littermates. (**h**) WAT from WT and miR-22^−/−^ mice fed an HFD for 8 weeks was stained for UCP-1; miR-22-deficient mice showed strong staining, suggesting brownization of WAT in miR-22 null mice. *, *p* < 0.05; **, *p* < 0.01; ***, *p* < 0.001; ****, *p* < 0.0001.

**Figure 2 ijms-24-12870-f002:**
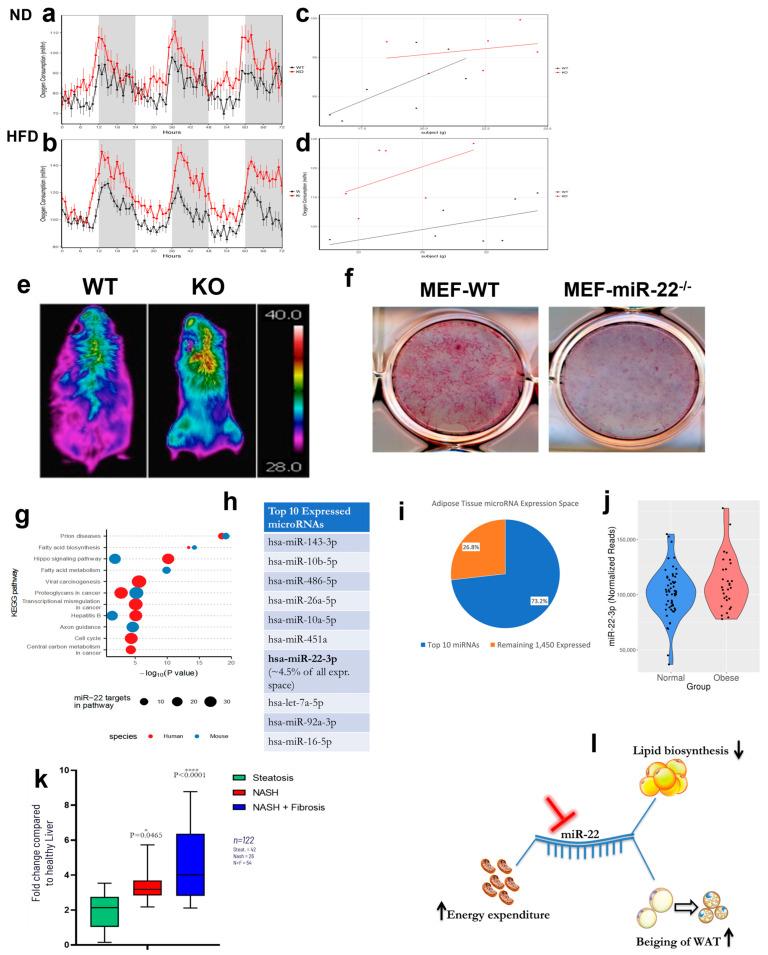
Genetic ablation of miR-22 prevents diet-induced obesity through metabolic rewiring. (**a**) Metabolic cage analysis shows that there is no difference in VO_2_ consumption between WT and miR-22 KO mice when fed a normal diet (*n* = 7). (**b**) Metabolic cage analysis shows that after 8 weeks on an HFD, KO mice have significantly higher VO_2_ consumption than WT mice despite lack of weight gain on the HFD (*n* = 7). (**c**) Regression analysis between VO_2_ and body mass from WT and KO mice fed with ND (*n* = 7). (**d**) Regression analysis between VO_2_ and body mass from WT and KO mice fed with an HFD for 8 weeks (*n* = 7)**.** (**e**) Representative thermal images of WT and miR-22^−/−^ mice after 8 weeks on an HFD. The miR-22^−/−^ mice are leaner than WT littermates, and they are characterized by a higher body temperature in the interscapular area, where BAT is located (*n* = 5 per cohort). (**f**) MEF cells from WT and miR-22^−/−^ mice were isolated and cultured in presence of adipose differentiation media for 8 days and then stained with Oil-Red-O. Staining of WT cells is much more intense than miR-22-deficient cells, implying that genetic ablation of miR-22 impairs adipocyte differentiation, making miR-22 null cells less prone to differentiate into adipocytes. (**g**) miRNA pathway analysis performed in humans and mice identifies pathways with overrepresentation of miR-22 target genes in TarBase and miR-22 targets in Reactome pathways. We used Diana-microT to predict miR-22 targets in humans and mice. Subsequently, we identified the number of predicted miR-22 targets annotated to different Reactome pathways. miR-22 is predicted to target genes across almost all major pathways in Reactome. (**h**) The 10 most highly expressed miRNAs in 185 human abdominal subcutaneous adipose tissue samples from the METSIM study [23562819]. (**i**) Pie chart depicting the number and percent (%) of next-generation sequencing reads assigned to the 10 most highly expressed miRNAs and the remaining 1450 miRNAs identified as expressed (>=1 sequencing read assigned) in 185 human abdominal subcutaneous adipose tissue samples from the METSIM study [23562819]. (**j**) miR-22 expression in human abdominal subcutaneous adipose tissue samples of individuals with BMI > 30 (*n* = 30) and BMI < 25 (*n* = 56) from the METSIM study [23562819]. Obese individuals exhibit increased miR-22 expression (*p*-value: 0.0078; q-value: 0.0373). (**k**) miR-22 level in different cohort of human patients diagnosticated with steatosis but not fibrosis, or F1-F2 fibrosis stage, or F3-F4 fibrosis stage. miR-22 level directly correlate with disease progression in human patients. (**l**) Proposed model of how miR-22 genetic inhibition impacts obesity and steatosis through different pathways that converge on the same metabolic advantage and protective effects.

## Data Availability

The data presented in this study are available on request from the last author. The data are not publicly available due to confidentiality related to patent application on going.

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
