# Peer review of "MicroRNA-22 Is a Key Regulator of Lipid and Metabolic Homeostasis"

_ijms, 2023, doi:10.3390/ijms241612870_

Round 1
Reviewer 1 Report
Dear Authors,
I commend you on your research investigating the role of microRNA-22 (miR-22) in regulating lipid and energy homeostasis and its implications for obesity and metabolic disorders. Your manuscript comprehensively analyzes the mechanisms underlying these conditions, offering valuable insights into potential therapeutic strategies.
The escalating prevalence of obesity poses a significant public health challenge, as it is associated with an increased risk of type 2 diabetes, cardiovascular disease, nonalcoholic fatty liver disease (NAFLD), and cancer. Identifying miR-22 as a crucial regulator in controlling obesity and hepatic steatosis is a major advancement. Using knockout and transgenic mouse models, you convincingly demonstrate the impact of miR-22 loss-of-function and overexpression on these phenotypes, even under normal dietary conditions.
Your study provides a detailed understanding of the mechanisms through which miR-22 affects lipid biogenesis and differentiation. By influencing multiple pathways associated with these processes, miR-22 emerges as a central player in energy metabolism regulation. Notably, your findings reveal that genetic ablation of miR-22 promotes metabolic rewiring towards increased energy expenditure and browning of white adipose tissue, suggesting the potential for miR-22 modulation as a therapeutic strategy for obesity and metabolic disorders.
The manuscript effectively presents your research objectives, methodology, and results, making it accessible to experts and non-experts. Additionally, your findings have implications beyond obesity, encompassing other metabolic disorders such as type 2 diabetes, cardiovascular disease, NAFLD, and cancer. This widens the scope of your research and underscores its relevance in the broader context of metabolic health.
Considering your work's quality and significance, I recommend accepting the manuscript without further modifications.
Author Response
We would like to thank Reviewer #1 for the positive evaluation of our manuscript and for recommending acceptance of the manuscript.
Reviewer 2 Report
The submitted manuscript investigates the role of miR-22 in lipid and metabolic homeostasis in mice. The study is well-designed and well-powered, and the manuscript is well-written and presented. However, the report is disingenuous as it is currently presented in that the authors do not present the study with appropriate reference to other publications in the field. Many of the results presented in the paper have been previously published and unreferenced in this paper. For example, a study published in 2020 (Thibonnier et al. BMJ Open Diabetes Res Care. 2020. doi: 10.1136/bmjdrc-2020-001478) showed that knockdown of miR-22 with an antagomir reduced bodyweight gain and adiposity (including hepatic adipose) and increased insulin sensitivity and glucose homeostasis by increasing energy expenditure without affecting food intake or activity. Studies such as this have not been cited by the authors, who instead present similar results as novel findings. This is a serious ethical issue, which may be accidental in this case, but nevertheless needs resolving. That said, there is sufficient interest in the study to recommend a major revision of the paper rather than rejection. The authors need to revisit the literature and decide exactly where the novelty in this study lies, and then rewrite the paper with due reference to the field.
Author Response
We would like to thank Reviewer #2 for objective review of our manuscript and appreciate the Reviewer’s comment on the lack of previously published work in the field.
We have revised the manuscript by adding the following sentences into the Introduction and Discussion sections: “These findings are consistent with the observed metabolic advantages upon pharmacologic inhibition of miR-22 in a mouse model of diet-induced obesity {Thibonnier, 2020 #967; Thibonnier, 2020 #924}. In the present study, we deployed miR-22 knockout and transgenic mice, respectively, and provide genetic evidence for an essential role of miR-22 in lipid accumulation, steatotic liver disease (STD) and obesity.”
“In this study, in the control of lipid and energy homeostasis as well as the onset and maintenance of obesity using miR-22 knockout and transgenic mice. These findings are consistent with the fact that miR-22 regulates direct mRNA targets implicated in several key metabolic pathways ( Thibonnier, 2020 #967; Thibonnier, 2020 #924).”
The data presented in our manuscript are in line with previous studies on miR-22. In our study, we provide key genetic evidence for miR-22 as an important rheostat controlling lipid homeostasis and the onset and maintenance of obesity, which is consistent with the fact that miR-22 regulates direct mRNA targets implicated in several key metabolic pathways (Thibonnier, 2020 #967; Thibonnier, 2020 #924). Importantly, we show for the first time that genetic deletion of miR-22 is able to protect mice from obesity under HFD conditions without affecting food consumption or lean mass further highlighting the fact that pharmacological inhibition of miR-22 function may represent a viable therapeutic strategy for the treatment of obesity and liver steatosis.
Reviewer 3 Report
In this study, Panella et al. explore the role of microRNA-22 in lipid and metabolic homeostasis and obesity. They have interesting observations; however, the authors need to address the following concerns for final acceptance.
Major concerns:
- One of the major concerns of the study is that so far, the authors have presented only observational studies, which makes it incomplete due to the lack of direct mechanistic investigation. Although the authors suggest that miR-22 controls multiple pathways related to lipid biogenesis and differentiation, there isn’t any evidence of direct molecular mechanisms of these outcomes. Experiments necessary for addressing the concerns:
- Identifying the mechanism of weight gain in the KO
- Mechanism of fat browning in the KO
- In vitro mechanistic studies with miR-22 silencing and overexpression on human adipocytes (primary or cell line would be acceptable)
Minor:
- The study needs a scholarly discussion where the results are discussed in a broader context, potential limitations are acknowledged, and suggestions for future research are made.
- The paper needs a clear conclusion section. A well-structured conclusion, summarizing the main findings, their implications, and potential future directions, would enhance the overall flow and comprehensibility of the paper.
- This manuscript must discuss any limitations of the study.
Author Response
We would like to thank Reviewer #3 for objective and thorough review of our manuscript.
We provided an extensive explanation with extra data to answer the Reviewer comments at our best. Please note that, as mentioned in the text, some of the data provided in the rebuttal letter to answer Reviewers comments are under strict confidentiality. We are happy to share those data but we kindly ask to handle them in the most careful way possible since there are pending patents and other publication that are based on these sensitive information.

Round 2
Reviewer 2 Report
The submitted manuscript investigates the role of miR-22 in lipid and metabolic homeostasis in mice. The study is well-designed and well-powered, and the manuscript is well-written and presented. The authors have revised the manuscript to include recent publications in the field and it now discusses the study appropriately.
Author Response
We are thankful for this reviewer comments and for their help in improving our manuscript
Reviewer 3 Report
The authors have responded to all the concerns satisfactorily. Since most of the studies are included in the upcoming manuscript, I recommend the authors to mention that in the discussion section citing these concerns as shortcomings which will be addressed in the future studies.
Author Response
We are thankful for this reviewer comments, we appreciate the fact that the data provided were helpful in resolving the reviewer’s doubts. To address the minor comment that was made in this second round of division we included in the discussion the following sentence
We also developed an anti-miR-22 therapy for obesity and hepatic lipid accumulation, that further validate our genetic models and that is helping us in better dissecting the molecular mechanism that is responsible for the observed phenotype. Data on tolerability and effectiveness of anti-miR-22 therapy for obesity will be the focus of a different manuscript that we are finalizing and that will also contains a more detailed mechanism that is a limitation of the present manuscript.
We hope that our action meets the reviewer expectation.